# The Hydroponic Rockwool Root Microbiome: Under Control or Underutilised?

**DOI:** 10.3390/microorganisms11040835

**Published:** 2023-03-24

**Authors:** Phil Thomas, Oliver G. G. Knox, Jeff R. Powell, Brian Sindel, Gal Winter

**Affiliations:** 1School of Science and Technology, University of New England, Armidale, NSW 2351, Australia; 2School of Environmental and Rural Science, University of New England, Armidale, NSW 2351, Australia; 3Hawkesbury Institute for the Environment, Western Sydney University, Penrith, NSW 2751, Australia

**Keywords:** hydroponic, horticulture, microbiome, microbial ecology, rhizosphere, rhizobiome, biocontrol, prebiotic

## Abstract

Land plants have an ancient and intimate relationship with microorganisms, which influences the composition of natural ecosystems and the performance of crops. Plants shape the microbiome around their roots by releasing organic nutrients into the soil. Hydroponic horticulture aims to protect crops from damaging soil-borne pathogens by replacing soil with an artificial growing medium, such as rockwool, an inert material made from molten rock spun into fibres. Microorganisms are generally considered a problem to be managed, to keep the glasshouse clean, but the hydroponic root microbiome assembles soon after planting and flourishes with the crop. Hence, microbe–plant interactions play out in an artificial environment that is quite unlike the soil in which they evolved. Plants in a near-ideal environment have little dependency on microbial partners, but our growing appreciation of the role of microbial communities is revealing opportunities to advance practices, especially in agriculture and human health. Hydroponic systems are especially well-suited to active management of the root microbiome because they allow complete control over the root zone environment; however, they receive much less attention than other host–microbiome interactions. Novel techniques for hydroponic horticulture can be identified by extending our understanding of the microbial ecology of this unique environment.

## 1. Introduction

The complex ecology of the root microbiome is the product of hundreds of millions of years of co-evolution between plants and microorganisms [1,2]. Plants promote the growth of beneficial microbes by releasing organic nutrients from their roots and can change these exudates in response to different conditions, hence influencing the composition of the root microbiome [3,4]. Some of the many microorganisms which have adapted to thrive in the soil rhizosphere, which has been conditioned by root exudates, are able to exploit this proximity further by invading tissue and causing disease [5,6]. Soilless horticulture was developed largely to address the problem of persistent soil-borne pathogens [7,8]. To enable plants to thrive without soil, hydroponic glasshouses provide a near-ideal environment while attempting to exclude other organisms. Rockwool, an absorbent, fibrous material made from volcanic rock, is a growing medium commonly used in commercial hydroponic systems in place of soil. By starting with this initially clean material and maintaining careful hygiene, the pathogen problem is reduced to a manageable level [8,9].

However, plants in hydroponic systems remain genetically primed to cultivate a healthy microbiome around their roots. Soon after planting, the release of organic nutrients enables microbial colonisation of the root zone [8,10,11], but rather than the vast diversity that roots encounter in soil, plants can only select microbial partners from those that manage to evade the measures meant to keep the environment clean. Rockwool provides an excellent substrate for anchoring roots while providing both moisture and air, but is otherwise unlike soil (Figure 1). Plant–microbe signalling mechanisms that have been calibrated by the selection of successful partnerships in soil may have quite different outcomes in the very different biotic and abiotic conditions of the glasshouse. The bulk flow of irrigation enables easy dispersal of microorganisms, especially in the direction of the flow, bringing into question the nature of the rhizosphere, an important focus of many microbe–plant and microbe–microbe interactions [5]. The ecology of the hydroponic rockwool root zone is therefore an expression of ancient plant-microbe relationships in a faux soil environment.

Providing an ideal, artificial environment for growth has also greatly diminished the importance of the root microbiome to plant health. Plants with ample nutrition do not benefit from microbial services such as the mobilisation of phosphorus or fixation of nitrogen, which can greatly affect fitness in soil. The glasshouse protects against environmental stress, and pests and diseases can be managed with well-established treatments. Non-pathogenic organisms that happen to cohabit with roots in the growing medium are seldom considered by growers because they do not influence crop performance. On the other hand, research into other host-associated microbiomes promises new opportunities to exploit our growing knowledge of these complex systems, such as precision medicine [12] and microbiome management for agriculture [13]. Unlike other agricultural systems, a hydroponic system provides direct and complete control of the root environment, making it remarkably well suited to active management of the microbiome. Moreover, the glasshouse environment is highly uniform, and growers strive for consistency throughout the crop, which is likely to give rise to a similarly consistent and stable crop-wide microbiome [14,15]. These characteristics encourage more exploration of methods to use the hydroponic microbiome to tackle real challenges in protected cropping.

## 2. Microbe-Plant Ecology

Land plants inhabited a microbial world even before they evolved roots and leaves [1,2,4,16,17,18] and as a result, have developed a dependency on microbial interaction for successful growth [19,20]. In natural environments, soil microbes regulate plant abundance, productivity, community structure, and diversity [20]. Microbes can benefit plants by suppressing pathogenic organisms, supplying or aiding in the acquisition of nutrients, releasing plant growth-promoting hormones [6], or activating the plant’s systemic immune system [21].

Conversely, plants can recruit beneficial microbes from the soil by releasing a range of metabolites such as organic sugars and acids [3,22], which can account for up to 40% of plant photosynthetic activity [6]. These root exudates, plus mucilage and detached cells, provide rich substrates for microbial growth, so the abundance of microorganisms in the root-affected soil (the rhizosphere) is much higher than that in the bulk soil [4,6]. Plant genotype, therefore, has a strong influence on the microbial composition of the rhizosphere and endosphere [19,23,24], and specific plants tend to attract a distinct group of microbial taxa, known as a core microbiome. Core microbiomes can be associated with a broad taxonomic group, such as a wide range of plant phyla [16], species [25], or even a varietal genotype within a species [19].

The range of microorganisms available for recruitment depends on the soil type and environment [24,26], but some soil microbes are better adapted to the root zone, so the rhizosphere usually develops a microbial ecology that is distinct from the bulk soil [3,4,22,27,28]. Microbial traits that provide a competitive advantage for survival (e.g., antimicrobial mechanisms, chemotaxis), colonisation of roots (e.g., attachment, biofilm formation), and interaction with plants (e.g., phytohormone production, nutrient mineralisation, elicitation of specific root metabolites [29]) are enriched in rhizosphere communities [23].

Microbe–microbe and microbe–plant interactions in the rhizosphere of plants in soil have been studied extensively in many natural and agricultural environments, under different conditions, and from different perspectives. Recently, the scope of rhizosphere research, and microbial ecology more broadly, has been expanded by the application of high-throughput sequencing, mass spectrometry, and bioinformatics technologies. Large-scale analysis of genetic markers, such as the 16S rRNA gene, has revealed the enormous taxonomic diversity of soil- and plant-associated microbiomes. However, our understanding of the principles that govern interactions in the root microbiome remains rudimentary, mainly due to the immense complexity of this environment in multiple, interrelated dimensions (e.g., ecological, genetic, spatial, temporal, and biochemical), the difficulty of observing individual interactions, and the lack of explanatory models.

## 3. The Hydroponic Root Zone

### 3.1. Hydroponic Horticulture

Soilless horticulture systems were adopted by commercial glasshouse growers in the late 1970s, primarily to protect crops from plant pathogens in soil [7,8]. Although they are much more expensive to establish than soil crops, hydroponic systems have advantages over horticulture in soil, such as independence from soil quality, more efficient use of land area, continuous crop production, reduced use of water, pesticides, and fertiliser, and avoiding depletion and erosion of natural soil [28]. However, growers soon discovered that completely excluding microorganisms from the glasshouse was not commercially feasible, and that the irrigation system made monoculture crops especially vulnerable to water-borne pathogens [8,10,11]. Starting with an effectively sterile growing medium also tends to deprive plants of the support and protection of natural microbial allies. However, hydroponics also enables closer control of inputs to the growing environments than in soil [8]. Consequently, methods of pathogen suppression in field crops were adapted and introduced into glasshouse hydroponic systems, and growers began to consider the introduction of beneficial microorganisms for pathogen control (biocontrol) and to promote plant growth (biostimulation) [30].

### 3.2. Hydroponic Rockwool as a Microbial Habitat

Rockwool is a manufactured growth substrate that performs many of the functions of soil but is effectively sterile prior to planting [7,8]. Like a loamy soil, it has an excellent capacity to absorb and retain water while also containing air pores for roots [7,9] (Table 1). Unlike soil, it is chemically inert, enabling complete control of nutrients provided to roots via the irrigation system [7,9]. During the day, the rockwool is regularly irrigated to field capacity with a solution of complete plant nutrients (fertigation) and then allowed to drain, creating a bulk flow of solution through the medium. Compared to soil, which offers an extremely varied environment of mineral and organic particles with different surface characteristics, grains and aggregates of different sizes and textures, and a wide range of soluble compounds, rockwool is a highly homogeneous environment [9].

Prior to planting a crop, the glasshouse surfaces and equipment are disinfected as much as possible to reduce the microbial load. The glasshouse growing environment (Figure 2) is physically contained, and workers and equipment are sanitised to minimise the risk of biological contamination of crop plants. Water used for irrigation, whether from an external source or recirculated from the crop, is treated by various antimicrobial methods, most commonly ultraviolet light, chlorination, ozonation, or heat. Most treatments are not specific to pathogens but tend to deplete all organisms and greatly reduce biological diversity; for example, ultraviolet and heat treatment can remove 99% of microorganisms [8,31]. Biological (slow) filtration is an alternative method of sterilisation that has been shown to reduce microbial load while maintaining diversity and to selectively reduce the abundance of water-borne phytopathogens [8,32] and phytopathogenic fungi [33].

The vigorous cleaning and control efforts greatly reduce the abundance of microorganisms in the growing area, but of course, it is not practicable to completely sterilise the glasshouse [8,11,34,35]. Once irrigation has been applied and plants have been installed, hydroponic systems are highly favourable environments for microbes: the rockwool growth medium is moist and nutrient rich; the air is humid and mild; the irrigation solution containing complete plant nutrition and root exudates is circulated in large volumes throughout the glasshouse. The abundance of bacteria, in particular, increases quickly as plant roots colonise the rockwool [8,10,11,34], reaching a steady density about a day after planting [14]. The daily cycle of solar input and fertigation drives patterns of plant activity (nutrient uptake, photosynthesis, and exudation), which govern the metabolic activity of most members of the microbiome. Microbes and plants simultaneously influence and respond to the chemical environment through exometabolites and root exudates, respectively.

The flow of irrigation through rockwool in a working hydroponic system produces vertical and horizontal gradients of pH and electrical conductivity (EC; ion concentration) in the slab [7]. The pattern of gradients is consistently related to the location of plants and irrigation drippers and is replicated throughout the glasshouse. Since these environmental factors have a strong influence on microbial community composition [36], this regular pattern (Figure 3) is likely to result in a corresponding regular variation in communities at predictable locations in the slab. The frequent flow of irrigation also enables the ready dispersal of microorganisms, especially in the direction of the flow.

Roots behave differently in hydroponic growing media; for example, Kamilova et al. [37] found that exudates of tomatoes were significantly greater in rockwool (45.60 µg/mg of dry plant weight) than in a neutral medium of glass beads (9.74 µg/mg). Rockwool favours the growth of bacteria more than fungi [38] and develops a different bacterial community from organic substrates under similar conditions [36]. The microbial community around the root, especially at the root tip, can influence the quality and quantity of exudates released [39], which in turn affects microbial growth.

### 3.3. Community Ecology in the Hydroponic Rhizobiome

Microbial colonisation of the hydroponic system is initially limited by a lack of organic nutrients [8]. Glasshouse hygiene measures hinder dispersal, so organisms with traits that enable access to the system are favoured. Colonisation can be achieved via several routes, for instance: by persisting during the cleaning process and the oligotrophic environment of the glasshouse before the installation of plants; by colonising seedlings in the nursery; by airborne dispersal via ventilation; by insect vector; by evading hygiene controls via equipment or worker apparel; and by water-borne dispersal via irrigation. Before planting, microbial abundance in the system is very low [14] and likely to consist of a small number of species able to survive the cleaning performed to prepare for the crop. This is in stark contrast to a natural soil environment, in which growing plants can recruit microbial partners from the most diverse reservoir of genomes on earth, and implies that the assembly of the hydroponic root microbiome is fundamentally different from soil [27]. Lack of diversity may limit functional redundancy, thereby reducing the stability of the microbial community [40].

Once plants are established, the root community is dominated by bacteria and maintains a consistent composition, which changes gradually as the growing season progresses [9,15,41,42,43], possibly associated with a shift in the metabolic profile [11], indicating that plant development is a strong determinant of community composition. For example, Rosberg et al. [41] found that the community composition was more strongly associated with plant age than the presence of the pathogen *Pythium ultimum*. Vargas et al. [42] found that the relative abundance of Paenibacillus and Flavobacterium decreased over a nine-month period while the proportion of unidentified bacteria increased.

Organisms that are adapted to both root interaction and the hydroponic environment have a competitive advantage. Root-attached (rhizoplane) microbial communities are consistently found to be differentiated from those of the circulating solution [15,28]. Edmonds et al. [28] found distinct differences between the root community of lettuce plants and the hydroponic solution from 12 days after germination, which continued to develop over time, suggesting that the root community is selectively recruited from the environment. As in soil, microbial diversity is lower in the rhizoplane than in the surrounding environment and lower again in the endosphere [19]. Highly adhesive bacteria have an advantage when first colonising living roots, and these strains can initiate a process of succession by facilitating colonisation by competitor strains [44]. This trait is likely to be more important in a hydroponic environment due to the substantial bulk movement of solution through the medium, compared with the normally very low rate of dispersion in soil. Root colonisation is an important capability for potential biocontrol organisms [45]. De Weert et al. [46] showed that chemotactic motility, the ability to move toward higher concentrations of root exudates, was important for tomato root colonisation by *Pseudomonas fluorescens*. Successful bacterial colonisers may adapt to the environment during the life of a single crop, enabling competitive specialisation even if the source species is reintroduced at planting. Norgaard et al. [47] showed that the common soil bacterium *Bacillus subtilis* could improve its fitness in a hydroponic rhizosphere through the selection of mutations associated with biofilm formation.

Plant genotype influences the selection of root-associated communities. French et al. [48] found that genotype accounted for 10% of the variation between root microbiota amongst six domesticated and two wild varieties of tomato, demonstrating that even closely related plants can select distinct root zone communities. Poudel et al. [19] found that microbial communities varied between different rootstocks in grafted tomatoes, i.e., different root genotypes developed distinctive microbiomes, independent of the scion (grafted stem of the plant), under the same conditions in a hydroponic system. Plant genotype may have a stronger effect in hydroponic systems, where roots provide nearly all organic carbon [9], than in soil environments, where many other factors influence the microbial environment [27].

Microbial communities in different locations but similar environmental conditions develop a similar composition, as observed in an experimental glasshouse system by comparing community fingerprints based on single-strand conformation polymorphism (SSCP) [15]. The glasshouse is highly uniform, and consistency of conditions throughout the crop is a major aim of growers. Gradients of moisture, pH, EC, and root exudates occur in regular patterns [7], resulting in the assembly of similar microbial communities at regular, predictable locations throughout the hydroponic system (Figure 4). For example, the zone below the plant stem and along the bottom of the slab, which is consistently moist and nutrient rich, can be expected to contain a different community than the zone between plants at the top of the slab.

The hydroponic microbiome is generally resilient to disturbance. Vallance et al. [15] found that the root community was altered by the application of the putatively beneficial organism *Pythium oligandrum*, but the oomycete did not persist on the root and the original community structure was re-established.

### 3.4. Microorganisms in the Hydroponic Root Zone

In previous studies of microorganisms in rockwool hydroponic systems, bacteria were not commonly identified to the genus or species level, except for those of special interest, such as the pathogens and beneficial organisms under examination. In irrigation solution, Berkelmann et al. [14] found *Pseudomonas* (40%), unclassified (18%), *Agrobacterium* (13%), *Xanthomonas* (9%), *Comamonas* (8%), *Azospirillum* (4%), *Enterobacter* (3%), *Flavobacterium*, *Alcaligenes*, *Rhodococcus*, *Yersina*, *Cytophaga*, and *Aureobacterium*.

Hydroponic systems are especially suitable for water-borne phytopathogens such as species of *Pythium* and *Phytophthora,* which produce zoospores that can spread rapidly through the irrigation system [8,11]. A very small inoculum of zoospores is sufficient to contaminate a hydroponic system [35]. The water content of the growing medium has been demonstrated to be the key factor affecting the density of water-borne pathogens in the system and the disease symptoms that they cause, such as Pythium root rot [8,49]. Khalil et al. [49] found that the load of *Pythium ultimum* on tomato plants in a climate chamber was 10-fold lower at 50% water content than at 70%. Infection is more common in recirculating systems, but converting to open irrigation (run-to-waste) does not affect established pathogens, as demonstrated in tomatoes on rockwool [10], indicating that these organisms can permanently colonise the root zone.

Pathogenic strains of *Fusarium* (e.g., *F. oxysporum* f.sp *radicis-lycopersici*) and *Pseudomonas* can also thrive in a glasshouse environment where there is little competition from their normal antagonists in soil [50]. Other pathogenic species commonly present in the roots of tomato in hydroponic systems are *Plectosphaerella cucumerina*, *Colletotrichum coccodes*, *Rhizoctonia solani*, *Verticillium nigrescens*, and *Verticillium albo-atrum* [51]. Plant infection depends on the density of the pathogen population, with low levels of potential pathogens often found in asymptomatic plants [51].

The absence of soil microorganisms is both a blessing and a curse. On the one hand, the removal of pathogens and replacement with beneficial microbes can improve crop performance [52]. On the other hand, the lack of evolutionary allies can hinder a plant’s defence, as demonstrated by Yin et al. [53], who found that tomatoes grown in sterile soil were more susceptible to bacterial wilt caused by *Ralstonia solanacearum*.

Root exudates have been linked to the presence of both pathogenic and beneficial organisms. Kamilova et al. [54] found that the pathogen *F. oxysporum* f. sp. *radicis-lycopersici* was associated with a decrease in citric acid and an increase in succinic acid, but the opposite was observed in the presence of *Pseudomonas fluorescens* biocontrol strain WCS365, along with a dramatic decrease in the proportion of diseased plants [54].

### 3.5. Biological Control

Much effort has been made to identify beneficial organisms that can be introduced to soil and soilless systems as biological control agents (BCAs) to suppress root diseases and promote plant growth [30,34,55,56]. Species of the ubiquitous soil fungal genus *Trichoderma* are often used for this purpose, due to their wide range of antifungal and plant growth-promoting capabilities [57]. Such treatments have often been found to be successful under experimental conditions [49]. Non-pathogenic *Fusarium* species, such as *F. equiseti,* have been found to control fungal diseases such as crown and root rot [58] and Fusarium wilt of tomato [59]. Bacterial BCAs are often found in the genera *Pseudomonas* [33,60] and *Bacillus* [8,30]. Phytopathogens can also be suppressed by the presence of an abundant commensal microbial community in the root zone [61,62]. For example, Tu et al. [62] found that the incidence of Pythium root rot was lower in a recirculating irrigation system compared with using a freshwater supply and estimated the optimal density of bacteria for disease suppression as 106 CFU/mL.

### 3.6. Limits of Current Understanding

The microbiology of hydroponic systems has been studied for more than 25 years [14], using methods that have advanced with the availability of new technologies. These include culture-based methods [62,63,64], including community-level physiological profiling (CLPP) [11,41]; molecular methods such as phospholipid fatty acid (PLFA) profiling [65,66]; and DNA-based technologies such as community fingerprinting via denaturing gradient gel electrophoresis (DGGE) [41,63]; and single-strand conformational polymorphism [11,15,67]. Currently, the predominant methods for characterising microbial ecology are DNA-based and include taxonomic barcoding, such as 16S rRNA amplification [61,63], and metagenomic analysis of environmental DNA [21,68].

Our understanding of plant–microbe interactions, along with other host-associated microbiomes, is now expanding rapidly with the development of powerful new molecular technologies, particularly high-throughput DNA sequencing and mass spectrometry, and a tremendous increase in the capacity and sophistication of bioinformatic analysis. These technologies have been applied successfully to elucidate complex microbial environments, such as soil [27] and activated sludge [69]. Species diversity and the effects of complex interactions have been identified as central factors influencing the outcome of microbe–plant relationships [17]. Both pathogenic [70] and beneficial [71] effects of the microbiome are more often linked to the consortia of microbes rather than single strains.

Hydroponic systems, however, attract much less attention than agricultural and natural soil environments, and the potential of these new technologies is not yet being realised. Hydroponic studies, including taxonomic barcoding, have often focused on individual pathogens or putatively beneficial microorganisms of interest or on identifying differences between communities without necessarily elaborating the underlying mechanisms to explain effects, thereby treating the microbiome as a “black box”, e.g., [55]. Surprisingly, it appears that no study of microorganisms in hydroponic systems has yet attempted to use metagenomic methods to profile novel species or examine the metabolic capability of the community.

Beneficial effects observed in lab conditions are often not reproduced in the field or glasshouse [8,45,71,72]. Many studies do not seek to replicate production glasshouse conditions, for example by including mature plants, e.g., [49,50,55,56] to examine microbiome interactions as root chemistry changes over the life of a crop [15,41] and plants become more susceptible to disease.

As a result, our understanding of the hydroponic root microbiome remains rudimentary. Basic questions remain unanswered, such as the importance of the microbiome to the crop, how to assess the health of the microbiome, the degree of variation between glasshouses or system designs, and whether novel organisms are present. The practical implications of transplanting the amazingly intricate relationship between roots and microbes into an artificial environment quite unlike the one in which it evolved are far from clear.

## 4. Opportunities

### 4.1. Biocontrol beyond the Root Zone

The biocontrol effect of the root microbiome is not limited to root pathogens but can extend to foliar pests via interaction with the plant immune system. Pineda et al. [73] demonstrated that rhizobiome manipulation could be used to include plant resistance against above-ground insect herbivores. Blundel et al. [74] found that the composition of the root microbiome was associated with levels of the phytohormones salicylic acid and jasmonic acid, which in turn increased the resistance of tomato plants to insect pests.

### 4.2. Crop Control

The growth of plants in soil can be improved by the presence of microorganisms that increase the availability of limiting nutrients, such as phosphorus [75], but this mechanism is not useful for hydroponically grown plants, which obtain complete nutrition from the fertigation solution. Growth promotion is still relevant to hydroponic systems, however. For example, biocontrol organisms (*Pseudomonas* and *Trichoderma* species) have been found to also promote tomato plant growth in rockwool hydroponic systems by reducing the burden of disease [34].

Crop steering, the regulation of plant development, especially the balance between vegetative and reproductive growth, is an important technique in commercial cropping operations, which is normally accomplished by variation of temperature and nutrition. Lu et al. [76] found that microbial communities can influence the development of *Arabidopsis thaliana* by changing levels of plant signalling molecules around the roots. Manipulation of the microbiome may therefore provide additional methods to coordinate production with harvest capacity and market demand.

Organisms in the microbiome can also influence the nutritional quality of fruit [77,78]. For example, inoculation with mycorrhizal fungi has been observed to increase levels of ascorbic acid and soluble sugars in tomatoes grown on rockwool [77]. Escobar Rodríguez et al. [78] found that the flavour and aroma chemistry of tomato fruit was associated with the relative abundance of bacteria in the root microbiome and that different phyla were associated with this effect in plants grown in soil or hydroponic systems.

Rootstock and scion varieties are fundamentally important to the success of a crop, and hence the subject of great interest to growers and plant breeders. However, the development of new plant varieties is time-consuming and expensive. De Palma et al. [79] found that 1243 transcripts were differentially expressed in tomato roots interacting with the plant-beneficial fungus *Trichoderma harzianum*. This suggests that the expression of the plant genome can be modulated by the application of a biocontrol organism, so the microbiome may provide an avenue for fine-tuning the performance of existing varieties.

### 4.3. Crop Monitoring

Plants stimulate the growth of beneficial microbes by the release of organic compounds from their roots and can alter their root chemistry in response to different conditions to influence the composition of the microbiome [3,22,29]. The microbial response to changes in root metabolism is likely to be more immediate and more sensitive in hydroponic systems because the plant provides virtually all the organic carbon in the root zone [9]. If microbial changes associated with plant responses to disease or stress can be detected before visible symptoms, then the microbiome can give early warning of problems in the crop, enabling early intervention to minimise impact.

Many examples of microbiome changes related to root pathogen infection in soil and soilless systems have been reported. For example, Larousse et al. [80] observed that root infection by the phytopathogenic oomocete *Phytophthora parasitica* resulted in changes to the microbial community composition of the rhizosphere of tomato plants in soil. Wei et al. [81] found that the relative abundance of beneficial bacteria was reduced when tomatoes in soil were infected by the root pathogen *Ralstonia solanacearum*. Conversely, Lee et al. [82] showed that manipulating the abundance of beneficial organisms induced the infection of tomatoes in soil by *R. solanacearum*, suggesting that the abundance of particular species could be used as an indicator of pathogen risk. Gu et al. [83] found that variation in microbiome composition was associated with the later incidence of disease in tomato plants in soil, also suggesting that the abundance of beneficial species can be an indicator of disease risk and also for planning preventative responses.

The root–microbiome response is also associated with above-ground events. Yuan et al. [84] demonstrated that the root microbiome of *Arabidopsis thaliana* in soil can be influenced by a foliar pathogen and that this was associated with changes in a range of root exudates. Adedayo et al. [85] found that the root microbiome of tomatoes infected with powdery mildew was different from that of healthy plants.

## 5. Steering the Microbiome

Hydroponic growers routinely manipulate nutrition and the glasshouse environment to steer the crop, for example, to regulate the balance between vegetative and reproductive growth. This capability also enables the microbiome to be steered via systematic manipulation of the root zone environment [9]. The growing medium is effectively sterile when the crop is planted [7], so primary colonisers do not face competition from established species. Unlike soil, there is no diverse reservoir of indigenous microorganisms and residual organic matter. The irrigation system is designed for direct and flexible control of the root environment (temperature, moisture, pH, EC, mineral nutrients, etc.), and growers strive for consistency throughout the crop, which leads to similar uniformity in microbial communities at regular locations in the system [15].

Antibiotic treatments (e.g., fungicides) and probiotic treatments, the application of biocontrol products, are routinely used to suppress or introduce microorganisms [9,30]. Biological treatments are becoming more preferred over chemical treatments due to their lower toxicity and wider range of purported benefits. To be effective, however, a biological agent must remain viable while being packed, shipped, stored in dry or liquid form, and finally applied to the root zone. Furthermore, the biocontrol organism(s) must compete with the incumbent root microbiome to become established and provide a benefit to the crop.

A major shortcoming of current treatments, which should be considered carefully when assessing other approaches, is our lack of knowledge of the ecology of the hydroponic microbiome. Biocontrol agents are managed in the same way as chemical treatments (product handling, preparation, application method, schedule, etc.). Recognising this process as an attempt to inoculate a new organism into the root microbiome raises questions that most growers are not equipped to answer. What is the current state of the microbiome? How is the biocontrol treatment expected to improve the microbiome? What does a healthy microbiome look like? How can the success of the treatment be assessed?

### 5.1. Synthetic Communities

Beneficial effects of the microbiome often arise from the consortia of microorganisms rather than from individuals [82,86]. Synthetic communities (SynComs) have been developed from groups of microbes either observed or predicted to confer desired plant benefits. While this approach is more technically advanced, the aims are the same for single-organism probiotics, and the SynCom must also be suited to the target environment and be able to compete effectively with successful microorganisms in the established community [87].

SynComs have been demonstrated to promote growth, increase plant immunity, and improve stress resistance [46]. For example, Tsolakidou et al. [75] designed two SynComs consisting of microorganisms obtained from compost that enhanced tomato growth and reduced disease symptoms. Schmitz et al. [88] demonstrated a SynCom that improved the tolerance of salt stress in tomato plants growing in a non-sterile substrate. Lee et al. [82] found that a community of four beneficial bacteria was more effective in activating plant immunity to *Ralstonia solanacearum* than each individual organism.

SynComs can reside within the plant root (the endosphere). By identifying biosynthetic gene clusters that were up-regulated in disease-suppressive soil bacteria, Carrión et al. [89] designed an endophytic consortium of seven plant-beneficial bacteria that suppressed disease caused by the fungal pathogen *Rhizoctonia solani*. Inoculating plant seeds with a beneficial SynCom has been suggested as a method for priming the endophytic community of young plants [86].

### 5.2. Prebiotic Biocontrol

The manufacture, transport, storage, and application of biological products are not straightforward. Products must generally be registered for use [9]. The inoculum must remain viable. As with chemical treatments, growers must comply with regulations governing food safety and the protection of the environment. A less logistically challenging way to manipulate the composition of the microbiome is through the use of prebiotics, nutrients that encourage the growth of desirable types of organisms already present in the microbiome. Ziazia et al. [90] found that the addition of general microbial growth media enhanced the resistance of eggplants to Verticillium wilt caused by *Verticillium dahlia*.

Prebiotic treatment can be likened to soil amendments, which are commonly used in soil crops [91], but this approach can be more flexible in a hydroponic system because soluble nutrients can be readily delivered to the root zone and varied over time, enabling the grower to steer the microbiome in a similar way to the crop. Moreover, the effect of the prebiotic is likely to be more precise because the range of organic material is much smaller and more consistent than in soil. Candidates for more sophisticated steering of the rhizobiome can be found in the array of primary and specialised metabolites released by the plants themselves [46].

### 5.3. Microbiome Breeding

Transplantation of the microbes extracted from mature plants has been suggested as a method for enhancing the microbiome of new crops [91] and has been demonstrated to suppress disease in tomatoes in soil [21]. This approach is difficult to apply in a hydroponic system, however, due to the risk of pathogens present in the inoculation material. Furthermore, the performance of some tomato varieties may be reduced by a tomato-conditioned microbiome [92].

Another method proposed for engineering the root microbiome is to selectively breed plants that attract beneficial microbes [93]. As with breeding for other characteristics, this approach involves the identification of plant quantitative trait loci that are associated with desired microbial taxa. Since it would add complexity and therefore cost to the already time-consuming process of developing a new commercial variety, this approach seems less flexible than other options and unlikely to be economically attractive to growers or plant breeders in the short term.

## 6. Conclusions

Like us, plants inhabit a microbial world. The microbiome of roots has a strong influence on the fitness of plants in natural systems and can make important contributions to the performance of crops in soil. In the hydroponic environment, however, the importance of the microbiome is greatly reduced, firstly by isolating plants from dangerous soil-borne pathogens, and also by removing the need for beneficial microbial services by directly supplying complete nutrition and a favourable growing environment. The irrigation system and growth medium are recognised as potential conduits for water-borne pathogens that need to be monitored and controlled, which growers need to monitor and control, but the benign majority of the microbiome is generally ignored. Biocontrol agents are favoured for the treatment of plant disease because of their low toxicity, but they are regarded simply as alternatives to chemical treatments rather than microbiome engineering. Most microbiological research in hydroponic systems to date has focused on the most economically damaging pathogens and the putative beneficial effects of biological products, while the ecology of the microbiome has received little attention.

The gap in our understanding can be seen by comparing the well-established horticultural practices used in most aspects of crop management to the methods used to monitor and manage the microbiome [9]. From scion and rootstock selection, environmental control, irrigation and nutrition, pest control and hygiene, to the labour force, market logistics, and environmental sustainability, growing practices are based on decades of research and development. However, the microbiome is literally invisible, and growers have little ability to assess its state or monitor change over time. As a result, the application of biological agents, and treatment of recalcitrant root pathogens, is often carried out by trial-and-error, which is much less effective and efficient than the precise and confident methods applied to other aspects of the crop.

Expanding our understanding of the hydroponic microbiome would clearly improve greenhouse practices, for example, by helping to manage the treatment of root pathogens, whether by chemical or biological methods. Moreover, the special capability to directly control the root environment suggests that there may be opportunities to apply emerging knowledge of host–microbiome interactions which are unique to hydroponic systems (Figure 5), such as sensitive crop steering and monitoring, stimulation of plant growth, and enhancement of plant health and fruit quality. However, glasshouse crops are presently only a small (albeit growing) proportion of agriculture, so there has been less economic incentive for suppliers to develop, test, and register hydroponic-specific products [9]. As a result, some currently available products are potentially unsuitable for a hydroponic environment, but perhaps the growing relevance of hydroponic cropping and high-quality produce will encourage the search for more effective organisms and consortia. While all agricultural systems consist of selectively bred organisms in an artificial ecosystem, the absence of soil in hydroponic systems is worthy of special attention because of the profound implications that it has for plant–microbiome interactions.

## Figures and Tables

**Figure 1 microorganisms-11-00835-f001:**
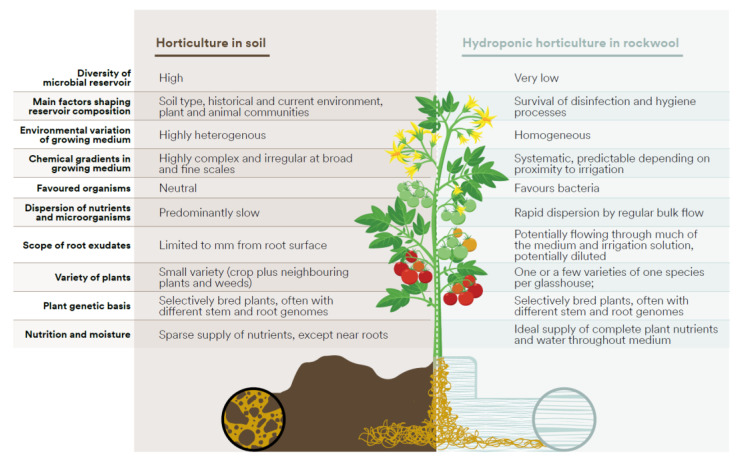
Comparison of broad features of soil and hydroponic rockwool environments in horticultural systems which affect the ecology of the root zone.

**Figure 2 microorganisms-11-00835-f002:**
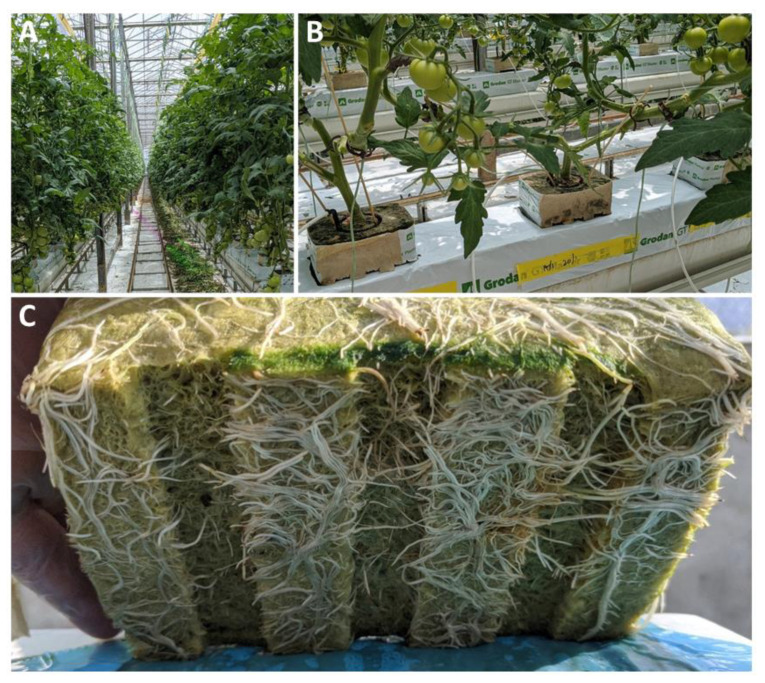
(**A**) Typical growing arrangement in a commercial glasshouse; (**B**) young tomato plants on rockwool slabs; (**C**) rockwool propagation block with roots of young tomato plant (images: Phil Thomas).

**Figure 3 microorganisms-11-00835-f003:**
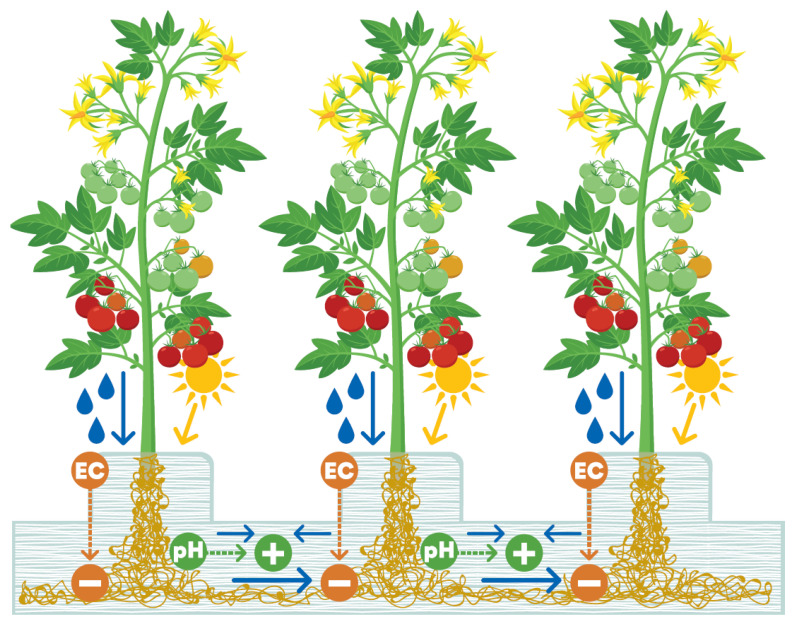
Rockwool growing medium as a microbial habitat, illustrating the flow of irrigation (blue arrows) and gradients of EC and pH, which are repeated throughout the glasshouse.

**Figure 4 microorganisms-11-00835-f004:**
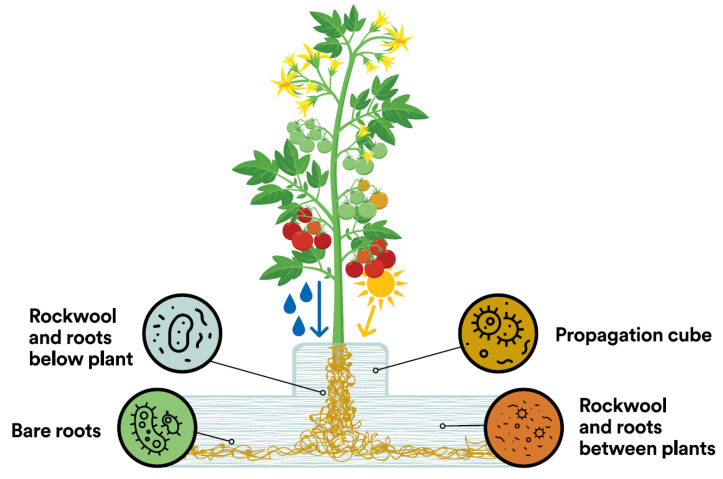
Assembly of distinct microbial communities at different locations in the root zone due to consistent variation in important environmental characteristics (e.g., pH, EC, sunlight, irrigation, root exudates).

**Figure 5 microorganisms-11-00835-f005:**
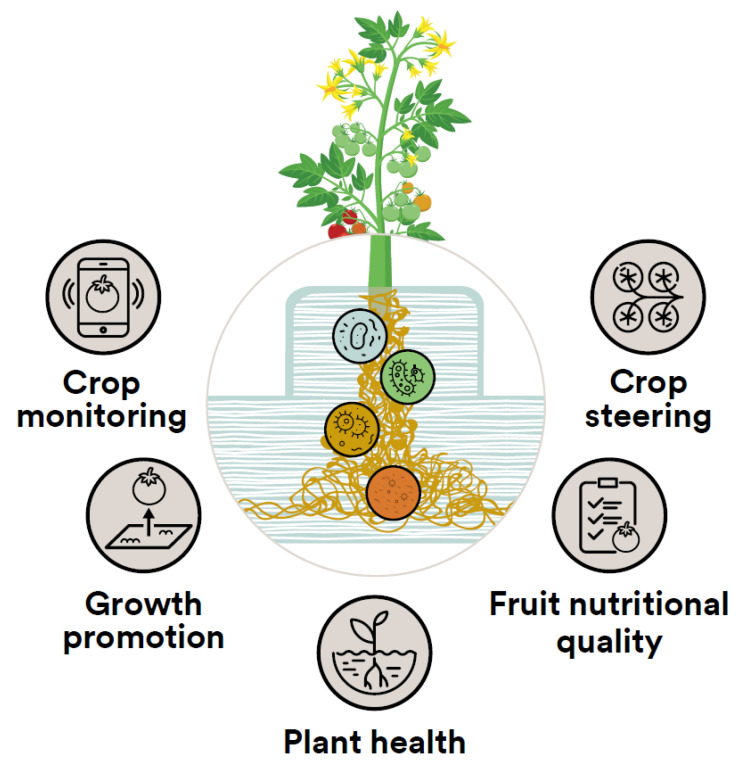
Potential applications of emerging host–microbiome research in hydroponic horticulture: crop monitoring and steering; plant growth promotion; and enhancement of plant health and fruit quality.

**Table 1 microorganisms-11-00835-t001:** Typical physical characteristics of common inorganic and organic hydroponic growing media [9].

Medium	Bulk Density (g/cm^3^)	Pore Volume (% by Volume)
Sand (0.02–2.0 mm particles)	1.48–1.80	30–45
Perlite	0.4–0.7	70–85
Rockwool	0.05–0.2	92–98
Sphagnum-based peats	41–263	84–96
Coconut coir	40–89	86–96

## Data Availability

No new data were created or analyzed in this study. Data sharing is not applicable to this article.

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
