# Peer review of "The Hydroponic Rockwool Root Microbiome: Under Control or Underutilised?"

_microorganisms, 2023, doi:10.3390/microorganisms11040835_

Round 1

Reviewer 1 Report

The manuscript is focused on the composition of a root microbiome in hydroponic systems and its variations under different environmental conditions. Particular attention has been paid to the advantages of rockwool as a substrate. The topic is very important and more information on the interrelation mechanisms between beneficial microorganisms and water-borne phytopathogens is urgently needed to bridge a knowledge gap in this field. Therefore, review papers on recent achievements and limitations in a root-microbiome sustainability are very appreciated. Authors collected and summarized a lot of recent publications on the topic. The manuscript is clearly written, well-structured. 

Some minor remarks (indicated in the file attached) are mostly related to the figures No., lack of specific data on i) rockwool properties and ii) shifts in root microbiome under particular conditions (which could be arranged in tables).

Kind regards

Reviewer 2 Report

Dear Authors and Editors,

This is a very interesting review.

(Row 48, the missing reference could be n°7. Line 142 does not need a biblio reference as several works are cited a few lines below and in a more specific and precise way).

I would not have been able to do such a precise and complete review on the subject. It should be published and will be very useful.

My opinion has relative value, then. I would still like to communicate it to the authors: since their review also shows that real soil seems to be the ideal substrate for plant growth (because it was born from the historical coevolution of plants and microorganisms), why continue to go in the direction of hydroponic solution?

I get it: hydroponic can be done anywhere, even vertically, and will be very useful for space travel.

But: plants and soil are the result of millions (counting only the last 500) of years of evolution (evidence of coexistence with feedback and reconstruction).

We know that hydroponic productions are the most modern simplification of what is called intensive agriculture. They are the most recent expression of the desire of humans to remake natural ecosystems (simplifying them to single elements, to understand and modify structure and functioning) to be able to handle them and feed a growing demand for food.

Personally, I would include a doubt in the conclusions: is this the right path for the development of our biological planet? I mean: if we Homo sapiens are among the most complex and intelligent organisms (apart from the microorganisms that we brought to the moon, I don't think other living beings have already managed to go up there) ever created by the evolution of natural ecosystems, why persist to believe that we can do better by ourselves? A doubt must come to the reader; the authors should promote this reflection, for ethical reasons and for the survival of our species. If we were born from natural ecosystems, why not promote already existing ecosystems (studying their complexity and understanding it for what it already is and produce) rather than rebuilding them in a way that we want only for ourselves?

A large-scale hydroponic production of food might not work, and could have disastrous returns for us. If it is not the individual species that evolve, but the ecosystems that contain them and the planet as a whole, creating simplified ecosystems could generate simplified products and other simplified ecosystems that are no longer suitable for a complex species like ours. It is well known that human pressure on ecosystems causes them to regress to earlier, more simplified evolutionary stages. On a planetary level, a generalized regression would take us back in time to periods when our species was not yet there. I don't know if it has already been demonstrated but it seems to me possible that the products (fruits and vegetables) of hydroponics can be less complex (chemically and physically) or less valid (with content in molecules linked to the complex functioning of the ecosystem as a whole) for the living who eat them of those produced naturally in a world with growing biodiversity.

In conclusion, could I suggest something useful to continue living on our planet: I would limit hydroponic experiments to laboratories, to understand the functioning of ecosystems, and I would not allow the generalization of their use for the production of food for the living beings of our planet. 

(I'm not an extremist, rather a normal university professor with family and children who study and are good; I have been vaccinated twice against covid-19 even if I have no intention of continuing to do so).

With compliments on the review and best wishes for your families and careers.
